# Natural Lagrangians

**Martin Tamm**

Department of Mathematics, University of Stockholm, 106 91 Stockholm, Sweden; matamm@math.su.se

**Abstract:** In this paper, a probabilistic approach is used to derive a kind of abstract candidate for a natural Lagrangian in general relativity. The methods are very general, and the result is in a certain sense unique. However, to turn this abstract Lagrangian into an ordinary one, expressible in terms of the Riemann tensor, is so far an open problem. Some possible cosmological consequences are discussed.

**Keywords:** general relativity; Lagrangian; ensemble; conservation of mass-energy

## 1. Introduction

Is there a "natural" Lagrangian in the theory of general relativity? In other words, does the concept of "action" have a fundamental meaning, or should we just consider Lagrangians to be technical tools that may be designed more or less at our convenience to meet the needs of the specific situation that we are studying?

Historically, there seems to have been a general belief that the action should be a natural and essentially unique concept. Still, when the Hilbert–Palatini principle ([1,2]) was formulated, starting from the simplest possible choice of action,

$$\delta \int R \, dV = 0, \tag{1}$$

this belief was quite common. However, already Eddington ([3]) was aware that many different Lagrangians would produce the same experimental predictions as the Hilbert–Palatini principle. Since that time, a large number of Lagrangians have been suggested, however without any particular one of them having been able to prove itself essentially better than all the others.

The answer that I will advocate in this paper is that there should be a natural Lagrangian. However, it may at the same time be that this Lagrangian can be something complicated and that the chances of finding it by guessing are very poor. Thus, we are led to the next question: Can we deduce the form of the natural Lagrangian? As it turns out, this may not be easy either.

Therefore, what should we do then? Let us start by briefly recalling how it all started.

## 2. The Principle of Least Action

The principle of least action has a long history. Often, it is attributed to Maupertius, who in the 18th Century formulated the belief that the universe develops according to a principle of ultimate economy (see [4]). However, already Leibniz had formulated something similar several decades earlier, and Fermat's principle (which can be viewed as a special case) is still older. Moreover, the mathematical formulation, stating that a physical system must develop in such a way that the action of the system is (locally) minimized, rather seems to be due to Euler and Lagrange.

That Maupertius is the one who is most closely associated with this principle may partly depend on the fact that he, more than others, emphasized the metaphysical aspects of the idea: to him, the fact that nature in a sense always chooses the most economical way of developing was the ultimate and indisputable proof of God's existence. However, not all

of Maupertius' contemporaries were convinced. Already Euler had noticed that the action of a system can sometimes in a sense instead be maximized, something that apparently does not fit too well with the idea of ultimate economy.

In spite of this, the principle of least action can only be regarded as a success story in science. However, at the same time, as this principle has developed from a metaphysical principle into an efficient instrument in any physicist's tool box, the philosophical aspects have so to speak faded away: not only would no scientist today take Maupertius' proof of God's existence seriously, but also, metaphysical ideas about nature's economy in general may seem somewhat out of date. Many physicists have come to the conclusion that questions about minimizing the action are essentially meaningless and that the best we can do is to look for stationary solutions of suitable Lagrangians.

However, maybe something got lost on the way? It may in fact appear somewhat unsatisfactory that the perhaps most universal principle of physics that we have is essentially a technical recipe; although references to nature's economy may seem unmotivated from a scientific point of view, the supremacy of stationary solutions in classical physics does not seem to have any acceptable motivation at all.

With the advent of quantum physics however, the principle of least action acquired a new meaning. Feynman's "democracy of all histories" approach to physics can in a sense be said to once more convert the principle of least action into a problem about optimization: the macroscopic developments that are actually realized are the ones that maximize the probability, and these turn out to be precisely those that are stationary with respect to the Lagrangian.

Part of the purpose of this paper is to suggest something similar for general relativity. The problem, however, is that we (in spite of numerous attempts) obviously do not know enough about the relationship between quantum mechanics and general relativity. For this reason, I will in this paper start by making use of a semi-classical probabilistic approach. This will permit us to formulate a kind of abstract form for a natural Lagrangian. There will however still be a non-trivial step to get from this abstract form to a Lagrangian in the usual sense (expressed as a natural function of the Riemann tensor).

### 3. The Random Curvature Ensemble

In this section, I consider a probabilistic approach to general relativity, which in a sense is a substitute for Feynman's democracy of all histories approach.

Thus, let us consider the probability space of all possible metrics on a certain space-time manifold, subject only to the condition that the total four-volume is a fixed number. Scalar curvature is essentially additive in separate regions. Therefore, what can we say about the probability for a certain value of the total scalar curvature in a region $D$ that is a union of many smaller regions?

For each such smaller region, we assume that there is a certain probability distribution for the different possible metrics. Exactly what this probability distribution looks like on the micro-level is of course difficult to know, but the point is that under quite general assumptions, this will not be important. Let us just suppose that it depends only on the scalar curvature. This is in fact very much in the spirit of the early theory of general relativity, where $R$ plays a central role, e.g., in the deduction of the field equations from the Hilbert–Palatini principle (compare [5]). We also suppose, starting from the idea that zero curvature is the most natural state, that the mean value of this distribution is zero. If we now consider the total curvature $R$ in $D$ to be the sum of the contributions from all the smaller subregions and if we (roughly) treat these contributions as independent variables, then the central limit theorem (see Fischer [6]) says that the probability for a certain value of $R$ is:

$$\sim \exp\{-\mu_\Delta R^2\}, \tag{2}$$

where $\mu_\Delta$ is a constant depending on the volume $\Delta$ of $D$. In the following, I will simply take this as the natural probability weight for the metric $g$ in $D$.

What about the probability weight $P$ of the metric $g$ on a larger set $U = \cup_\alpha D_\alpha$? Assuming multiplicativity (which means that different regions are treated as essentially independent of each other) and that all the regions have roughly the same volume $\Delta$, we get the (unnormalized) probability:

$$P \sim \prod_\alpha \exp\{-\mu_\Delta R_{g_\alpha}^2\} = \exp\{-\mu_\Delta \sum_\alpha R_{g_\alpha}^2\} \approx \exp\{-\mu \int_U R_g^2 \, dV\}. \tag{3}$$

Here we have, in the transition from sum to integral, made use of the additive property of the variance in normal distributions, which in this context means that $\mu_\Delta \approx \mu\Delta$ for some fixed constant $\mu$. Therefore, what we obtain is a kind of ensemble of all possible metrics in $\Omega$, where each metric gets a probability weight as above.

The word ensemble originally comes from statistical mechanics, and the idea is now to apply methods from classical statistical mechanics to the probability space of all metrics (see, e.g., Huang [7] for some background about ensembles).

First, compute the "state sum":

$$\Xi = \sum_g \exp\{-\int_U \mu R_g^2 \, dV\}. \tag{4}$$

Minus the logarithm of the state sum, $L = -\log \Xi$, is what is usually referred to as the "Helmholtz free energy". According to standard wisdom in statistical mechanics, the macrostates that minimize $L$ (among all states with a given volume) are the by far the most probable ones, i.e., the ones that will be realized in practice.

**Remark 1.** *Note that these ideas are here applied to four-dimensional states, not to three-dimensional ones as in usual statistical mechanics. In particular, the Helmholtz free energy is not directly connected to ordinary energy. Rather, I use the term here to relate to a very general statistical principle and a traditional way of thinking.*

Therefore, what does the free energy $L$ look like in our case? Again according to standard wisdom in statistical mechanics, the sum in (4) above is usually dominated by its largest term together with all terms corresponding to nearby metrics. The number of such metrics $g + \delta g$ near a given metric $g$ (which give approximately the same value of $R$) defines what is usually called the "density-of-states". If we let $\Omega_\alpha$ denote the number of such states in each set $D_\alpha$ as above, then we can heuristically compute the state sum,

$$\Xi = \sum_g \prod_\alpha \exp\{-\mu_\Delta R_{g_\alpha}^2\} \approx \sum_{\delta g} \prod_\alpha \exp\{-\mu_\Delta R_{g_\alpha}^2\}, \tag{5}$$

in the following way. First, note that according to the very definition of the "density-of-states", for all terms in the sum in (5) that significantly contribute, the exponential factors will be essentially the same. If we in addition suppose that the metric $g + \delta g$ can be viewed as given by an independent choice $\delta g_\alpha$ in each $D_\alpha$, then $\Xi$ can formally be rewritten as:

$$\Xi \approx \sum_{\delta g} \prod_\alpha \exp\{-\mu_\Delta R_{g_\alpha}^2\} \approx \prod_\alpha (\underbrace{1 + 1 + \ldots + 1}_{\Omega_\alpha \text{ terms}}) \exp\{-\mu_\Delta R_{g_\alpha}^2\} \tag{6}$$

$$= \prod_\alpha \Omega_\alpha \exp\{-\mu_\Delta R_{g_\alpha}^2\} = \prod_\alpha \exp\{-\mu_\Delta R_{g_\alpha}^2 + \log \Omega_\alpha\}. \tag{7}$$

Writing as before $\mu_\Delta = \mu\Delta$, we note that the approximate independence of the cells $D_\alpha$ means that $\Omega_\alpha$ is an exponential function of the volume of $D_\alpha$. Hence, it is natural to write $\log \Omega_\alpha = \log \Omega_g \Delta V$, where $\log \Omega_g$ is now a measure of the density-of-states of $g$ itself, which is essentially independent of the particular choices of the $D_\alpha$s.

Summing up, after a transition to an integral as in (3), we arrive at:

$$\Xi \approx \exp\left\{-\int_U \left(\mu R_g^2 - \log \Omega_g\right) dV\right\}, \tag{8}$$

or equivalently:

$$L = -\log(\Xi) \approx \int_U \left(\mu R_g^2 - \log \Omega_g\right) dV. \tag{9}$$

The principle of minimizing the free energy now gives us a natural, although of course still heuristic, foundation for the following.

**Principle 1** (Of least action). *The metric g that is realized in U must minimize:*

$$L = \int_U \left(\mu R_g^2 - \log \Omega_g\right) dV, \quad (Vol(U) \ fixed). \tag{10}$$

In general, finding the states that minimize the free energy can be very difficult, since they are determined by a sensitive interplay between the size of the terms in the state sum and the corresponding density of the state function. In the present situation, however, this difficulty may be overshadowed by a still more difficult problem: How do we compute $\Omega_g$? This is a very non-trivial problem in infinite-dimensional differential geometry. In fact, there is also the problem of how the density-of-states function should be defined. To the mind of the author and again referring to standard methods in statistical mechanics, this second problem may be less serious, since in statistical mechanics, the consequences are usually very insensitive to the details in the exact definition of $\Omega$.

Before we consider these questions in general, let us in the next section first consider the simplest case.

## 4. The Field Equations in a Vacuum

Clearly, the very least one must ask from a Lagrangian is that it should be able to reproduce the field equations in a vacuum. In general, the metric that minimizes the action integral in (10) is determined by the interplay between the $R^2$ term and the density-of-states term $\log \Omega_g$. However, if the general situation and the boundary conditions allow for a metric $g$ that satisfies $Ricci = 0$, one can argue that this metric must also be minimizing. The reason is that such a metric will in fact simultaneously minimize both terms in (10). For the first term, this is obvious: clearly, $Ricci = 0$ implies that $R = 0$, which of course minimizes the $R^2$ term. For the second term, the reason is more subtle, and I can here only give a heuristic argument. From statistical mechanics, it is well known that $-\log \Omega$ will be minimized ($\log \Omega$ will be maximized) when the minimum of:

$$\int_U R_g^2 \, dV \tag{11}$$

is as "flat" as possible, i.e., in this case when the derivatives of $R_g$ (with respect to various directions in the space of all metrics) are as small as possible.

Consider therefore a differentiable one-parameter family $g_s = g_0 + s \cdot h$ of metrics passing through a given extremal metric $g = g_0$. Computing the $s$-derivative of the scalar curvature $R(s)$ along this one-parameter family gives that:

$$\frac{dR}{ds} = -\sum_{i,j} h^{ij} R_{ij} + \text{divergence terms}, \quad \text{where} \quad h^{ij} = \sum_{k,l} g^{ik} g^{jl} h_{kl} \tag{12}$$

(see, e.g., [8]. For a more complete, but also more difficult to read, account, see [9] or [5]). In the present context, the scalar curvature is always integrated. If we consider variations $h$

with support small enough to be contained in the domain of integration, the divergence terms will disappear when integrated, and we will be left with:

$$\frac{dR}{ds} = -\sum_{i,j} h^{ij} R_{ij}. \tag{13}$$

It is clear that if the minimum is flat in all directions, in the sense that the left-hand side in (13) vanishes in all directions, then $Ricci = 0$, since if the sum on the right-hand side vanishes for all possible choices of $h^{ij}$, then also all $R_{ij}$ must vanish. Put into other words, this would mean that $-\log \Omega$ should be minimized exactly when the vacuum field equations are fulfilled.

### 5. Towards the General Classical Case

It is not always the case that the situation allows for $Ricci = 0$ to be fulfilled. This could be for global or topological reasons, but most commonly simply because there is matter present. Therefore, what can be said in this case?

The first thing to observe is that according to general relativity, mass will affect the first term in (10). In fact, any body with mass should give rise to a nontrivial Schwarzschild metric far away from it. However, closer to the particle, such a metric will then by necessity have to have a non-zero Ricci tensor and in general also a non-zero scalar curvature.

**Remark 2.** *It can of course be argued that this may not be true if we allow for singularities, like, e.g., in the Schwarzschild metric. However, if we want the theory to allow for singularities, we must at the same time find a consistent way of giving sense to and computing generalized integrals of the curvature tensor at such points. This may be very difficult, so for the time being, all metrics will be assumed to be non-singular.*

It is much more difficult to say what will happen to the density-of-states term. A direct computation would have to be carried out in the infinite-dimensional space of all metrics. So far, we have not even defined any specific differential structure on this space, and there are certainly many possible choices for such a structure. Even if it seems reasonable to expect the minimizing metric $g$ to be essentially independent of this choice, nevertheless, this appears to be an extremely difficult problem.

There may however be another way to proceed. Let us start by asking ourselves what properties $-\log \Omega_g$ should have. First of all, let us observe that it should be an essentially invariant, locally-defined concept, which should only depend on the geometric properties of $g$. Thus, it seems reasonable to expect it to be expressible in terms of the Riemann tensor. Moreover, according to the discussion in the previous section, it seems very natural to suppose that this tensor should in a vacuum be locally minimized precisely when $Ricci = 0$.

Therefore, instead of trying to compute $-\log \Omega_g$ directly, we could start by asking if these properties determine this tensor more or less uniquely, or if this is not the case, at least reduce the number of possible candidates to a small number. Even if it may still be difficult to know exactly what constraints the presence of mass puts on the metrics (except that they should reduce to the ordinary field equations on a large scale), I would like to suggest this as an interesting open problem:

**Problem 1.** *Find and characterize all tensorial densities that are locally minimized exactly when $Ricci = 0$.*

A solution to this problem may not give us the final form of the natural Lagrangian, but it could in fact be a major step towards it. Unfortunately, little seems to be known about this. In fact, it does not even seem to be known if such a tensor exists at all. Nevertheless, this essentially algebraic problem seems to be easier to work with than the direct computation method.

**Remark 3.** *In principle, it is of course possible to relax the condition that minimization should be equivalent to Ricci = 0, by saying only that Ricci = 0 implies minimization. Such tensors are easier to find (a trivial example being $(R_{ij}R^{ij})^2$). However, it is not clear what kind of physics this would lead to, so equivalence would seem desirable.*

## 6. The Concept of Mass-Energy

An early idea of the meaning of action can be said to have been the integral:

$$L = \int E\, dt, \tag{14}$$

typically along the path of a certain body or, more generally, along the paths of several bodies. Although the interpretation of this concept may have changed somewhat during the development of modern physics, it can be interesting to see what (14) would lead to if combined with the action in (10).

In fact, reversing the original idea implicit in (14), we are led to the following:

**Definition 1.** *The total mass-energy of a system, as measured during the time interval $[T_1, T_2]$ and in a certain region U in space, can be computed as:*

$$E = \frac{1}{T_2 - T_1} \int_{x \in U, T_1 \le t \le T_2} \left( \mu R_g^2 - \log \Omega_g \right) dV. \tag{15}$$

As it stands, this energy also contains vacuum energy: even if $R = 0$ in a vacuum, the second term will in general be expected to be non-zero. However, in most ordinary physical applications, we do not want to include this vacuum energy. In particular, we can study the case of a close to flat space-time with the constant vacuum density $-\log \Omega_0$ subtracted away.

**Definition 2.** *In the case of a single isolated particle at rest in the given frame of reference, Formula (15) can be used to define its rest mass in terms of curvature alone as:*

$$M = \frac{1}{T_2 - T_1} \int_{T_1 \le t \le T_2} \left( \mu R_g^2 - \log \Omega_g + \log \Omega_0 \right) dV = \tag{16}$$

$$\frac{1}{T_2 - T_1} \int_{T_1 \le t \le T_2} \left( \mu R_g^2 - \log(\Omega_g / \Omega_0) \right) dV. \tag{17}$$

For this to make sense, it is of course necessary to suppose that whatever influence the particle may have on its surroundings far away from the particle, the contribution to the integrand in (16) will be negligible. For the rest of this section, it will be assumed that the vacuum density $-\log \Omega_0$ has been subtracted away from the action as in (17).

Is this a reasonable definition of mass? First of all, it should be kept in mind that we are here only concerned with gravitation: whatever other forces could contribute is so to speak left out from the beginning. Having said this, there are still many questions that have to be answered, for example:

I.   Will particles travel along lines or, more generally, along geodesics? This can hopefully be proven true at least for suitable classes of particles; however, it is not obvious from the above, and the question turns out to be surprisingly complex. Part of the reason for this is that although geodesics are stationary and from the point of view of the democracy of all histories should be probability maximizing, there is also the $R^2$-term in (17). This term in general tends to be proportional to the length of the path, which means that it rather tends to maximize the action along geodesics, since in general relativity, these tend to have maximal length. Therefore, will the result still minimize the action (maximize the probability)?

It is not possible to estimate the relative importance of the two terms in general without additional information. In the opinion of the author, it is quite possible that one can construct examples of situations where the answer is no. On the other hand, however, it is also quite possible that for certain classes of metrics, the answer will be yes.

II.  Will mass energy be conserved? Assuming a positive answer to I. above, the answer in this case will in a certain sense also be yes. However, in situations with very high curvature or that, e.g., concern the universe as a whole, it may be that the usual idea of conservation must be modified (see Section 7). However, let us start by considering the most common situation in an essentially flat setting.

**Principle 2** (Conservation of mass-energy). *Consider a collection of particles in some region in space-time where the deviation of the geometry from flat space-time is negligible except in the immediate vicinity of the particles. We assume that they move independently along straight lines except that they may momentarily interact by emission and absorption of (real or virtual) particles, thus changing their states of motion and other properties. If we define the total action as the sum of all actions along the world-lines of all the involved particles, then the principle of least action implies that the usual energy of the system, defined as:*

$$E = \sum_{i=1}^{n} \frac{M_i}{\sqrt{1 - u_i^2}},\tag{18}$$

*is a conserved quantity, where $M_i$:s denote the masses of the particles as in (17) at the given time and $u_i$:s denote the speeds of the corresponding particles.*

To motivate this principle, we first note that the contribution of a particle with rest mass M to the action, during a time interval of length $\Delta t$ where it is not interacting with anything, can be written as $MT$, where $T = \sqrt{\Delta t^2 - |\Delta x|^2}$ is the proper time elapsed.

Now, consider the mass-energy $E(t)$ as measured during a time interval $[t, t + \Delta t]$ of length $\Delta t$, where no interaction takes place. Then:

$$E(t) = \frac{\Delta \Xi}{\Delta t},\tag{19}$$

where $\Delta \Xi$ is the sum of the actions of the individual particles computed in the time-interval $[t, t + \Delta t]$. It is now claimed that $E(t)$ must be constant as a function of $t$, because otherwise, one can easily construct a volume preserving infinitesimal transformation, which decreases the action by contracting the time scale at some time $t'$ where $E(t)$ is large and simultaneously expanding the time scale at some other time $t''$ where $E(t)$ is smaller.

**Remark 4.** *Although this is a standard idea in statistical mechanics, it is worth pointing out that the argument depends heavily on the fact that the metric is supposed to be minimizing along the paths of the particles. In fact, an infinitesimal deformation as above will bend the path, which may effect the curvature. However, exactly because the metric is supposed to be minimizing along the straight line (geodesic), this contribution will be of second order, hence be negligible in comparison with the first order contribution resulting from the changes in the time scales.*

We conclude that the time-derivate of the total action must be constant. Hence, with:

$$\Xi = \sum_{i=1}^{n} M_i T_i = \sum_{i=1}^{n} M_i \sqrt{\Delta t^2 - |\Delta x_i|^2},\tag{20}$$

we can now compute the conserved quantity as:

$$E(t) = \frac{\partial}{\partial t}\Xi = \sum_{i=1}^{n} M_i \frac{\Delta t}{\sqrt{\Delta t^2 - |\Delta x_i|^2}} = \sum_{i=1}^{n} \frac{M_i}{\sqrt{1 - u_i^2}} \tag{21}$$

where:

$$u_i = \frac{|\Delta x_i|}{\Delta t} \tag{22}$$

is the speed of the $i$th particle. Thus, the claim follows.

**Remark 5.** *Here, we only consider energy conservation. However, since everything is obviously Lorentz invariant, we must have a similar conservation law for the momentum.*

### 7. Is Mass-Energy Conserved Globally?

Is mass-energy always conserved in general relativity? Although we may in general want the answer to be yes, the question is more complicated than it may seem to be at first sight (compare, e.g., [10]). In particular, there are situations on the global scale where the behavior that we observe seems to contradict conservation. As an example, one may ask what happens to the negative potential energy between galaxies when they drift apart with accelerating speed?

The definition of mass-energy in (15) offers a possible explanation: In usual general relativity, the relation between the mass-energy of particles and the underlying geometry is somewhat asymmetric. However, here, they appear on a more equal footing. In fact, the influence of both comes from their contribution to the integral:

$$\int_{\Omega} \left( \mu R_g^2 - \log \Omega_g \right) dV, \tag{23}$$

and in the same way and on equal terms. In other words, the global curvature of space-time will also contribute and can hence be considered as a kind of geometric energy.

Thus, the total conserved quantity $E$ may be considered to consist of two parts:

$$E = E_{\text{mass-energy}} + E_{\text{global geometry}}. \tag{24}$$

This may give a more fundamental version of the conservation law for mass/energy. For cosmology, this means that each one of the two terms need not be conserved, only the sum.

To illustrate this, let us consider an extremely simple model for a closed, homogeneous, isotropic universe with a given fixed volume, where we neglect all physics except the part that comes from gravitation and curvature. The metric of such a universe can be written as:

$$ds^2 = -dt^2 + a(t)^2 \left( d\chi^2 + \sin^2 \chi (d\theta^2 + \sin^2 \theta d\varphi^2) \right). \tag{25}$$

Since we are dealing with a closed model, the function $a(t)$ can naturally be thought of as the radius of the universe at time $t$. What form will such a universe have if we start from the principle of least action as in Section 3? This is a non-trivial question due to the difficulties in calculating the density-of-states term. If we, however, for the sake of argument, temporarily leave out this term and concentrate on the scalar curvature, then the action can be computed as follows.

Computing the scalar curvature of (25) gives:

$$R(t) = \frac{6(1 + a'(t)^2 + a(t)a''(t))}{a(t)^2}. \tag{26}$$

From this, we obtain, in view of the assumption about isotropy and during a certain interval of time $I$ (which essentially could be the lifespan of the universe),

$$\int_\Omega R^2 \, dV = 2\pi^2 \int_I R(t)^2 a(t)^3 \, dt = 2\pi^2 \int_I \frac{36(1 + a'(t)^2 + a(t)a''(t))^2}{a(t)} \, dt. \tag{27}$$

In the case of no mass/energy, it is easy to see that this integral will be minimized when the universe is a four-sphere. In fact, the scalar curvature in this case is identically zero. However, what happens if we add to the action an energy term corresponding to a homogeneous mass distribution? If the amount of mass does not change with time, then the mass itself will not influence the minimizing of the action. What will influence it however is the potential energy. Taking this energy into account in a completely classical way amounts to adding in (24) a term that is inversely proportional to the radius of the universe at the given time, we get the following expression for the total action:

$$L = 2\pi^2 \int_I \frac{36(1 + a'(t)^2 + a(t)a''(t))^2}{a(t)} \, dt - 2\pi^2 \beta \int_I \frac{1}{a(t)} \, dt, \tag{28}$$

for some constant $\beta > 0$. The potential energy term in (28) can be thought of as a kind of semi-classical substitute for the density-of-states term $\log \Omega$. In a more complete theory, one could hopefully do without references to such concepts as potential energy. However, we are not there yet.

Minimizing this expression under the condition of a constant volume $V$, according to the classical theory of the calculus of variation, means computing the Euler–Lagrange equation for the functional:

$$\Phi = 2\pi^2 \int_I \frac{36(1 + a'(t)^2 + a(t)a''(t))^2}{a(t)} \, dt - 2\pi^2 \beta \int_I \frac{1}{a(t)} \, dt + \lambda \left( V - 2\pi^2 \int_I a(t)^3 dt \right). \tag{29}$$

For a general functional of the form:

$$\int_I F(a(t), a'(t), a''(t)) \, dt, \tag{30}$$

where $F(u, v, w)$ is some sufficiently regular function, the Euler–Lagrange equation (see [11]) is given by:

$$\frac{\partial F}{\partial u}(a(t), a'(t), a''(t)) - \frac{d}{dt}\left( \frac{\partial F}{\partial v}(a(t), a'(t), a''(t)) \right) + \frac{d^2}{dt^2}\left( \frac{\partial F}{\partial w}(a(t), a'(t), a''(t)) \right) = 0. \tag{31}$$

Using Mathematica, we can now compute the Euler–Lagrange equation associated with the functional in (28) and obtain (after multiplying with $a(t)^2/2\pi^2$):

$$2a(t)^3 a^{(4)}(t) + 3a(t)^2 a''(t)^2 - 4a(t)a''(t) + 3a'(t)^4 + 2a'(t)^2 +$$

$$4a(t)^2 a^{(3)}(t)a'(t) - 12a(t)a'(t)^2 a''(t) - 1 + \beta = 3\lambda a(t)^4, \tag{32}$$

Analyzing all solutions of this equation for all choices of the parameters is a huge task, which I will not attempt here (see [12] for more technical details about these computations). Just as an example, I used Mathematica to plot one more or less typical solution with $\beta = 1.5$ and $\lambda = 1.525$ and with a phase of accelerating expansion in Figure 1.

**Remark 6.** *It should be noted that the model here does not contain enough assumptions to give a realistic picture of the behavior close to the Big Bang. In particular, it is not possible to solve the Euler–Lagrange equation starting from a point where $a(t)$ is zero, without additional assumptions. Rather, the solution in Figure 1 was obtained starting from the point where $a(t)$ is maximal.*

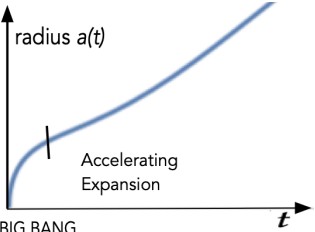

**Figure 1.** A more or less typical solution to the Euler–Lagrange equation.

In this model, the scalar curvature decreases during the expansion phase, as can be seen by inserting the solution of (32) into (27). Hence, it is only natural that the mass-energy part grows during the same period of time according to the formula:

$$E_{\text{mass-energy}} = E - E_{\text{global geometry}}. \tag{33}$$

Computing $E_{\text{global geometry}}$ in the above model (neglecting the density-of-states term) leads to a qualitative picture as in Figure 2 for the mass-energy that we may observe. Again, the behavior close to the ends may be very inaccurate.

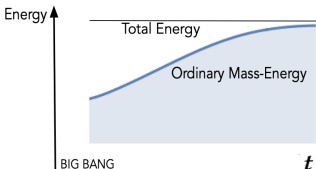

**Figure 2.** A plot showing the growth of the ordinary mass-energy during the expansion phase.

**Remark 7.** *The exclusion of the density-of-states term significantly simplifies the computations in this section. However, how realistic is this assumption? To give a definite answer to this question definitely requires more research.*

*However, it can be noted, as in Section 4, that in the case of the vacuum, the scalar curvature term and the density-of-states term both tend to be minimized simultaneously, which at least makes it plausible that for low curvature, these two terms will tend to have similar growth properties in various directions. This might indicate that the Euler–Lagrange equation for their sum could be similar to the one studied above.*

## 8. Conclusions

The model in the previous section does of course not claim to give an exact picture to be compared with experimental data, but rather aims at an explanation for the underlying mechanism for the conservation and possible non-conservation of mass/energy.

In general, this paper is clearly only a preliminary attempt to formulate a natural Lagrangian for general relativity. It would of course be very interesting to replace the essentially classical ensemble in Section 3 by a quantum mechanical one. This may not be impossible, but there are still many technical problems that have to be solved on the way.

One may also ask what the connection is between the ideas in this paper and other current areas of research, like extended theories of gravity and in particular $f(R)$ gravity theories ([13–16]). Clearly, there is an obvious common goal, but the approaches are quite different. The main stream in physics has always been to suggest explicit theories and then test them against reality. In general, the method has been enormously successful, but the success depends on our ability to launch the right candidates. The starting point for this paper is the opposite one: it may be very difficult to find the right candidate without previously having reduced the number of possibilities drastically. However, this way of approaching the problem has other drawbacks, and in particular, it seems to lead to extremely difficult mathematical problems.

In the mind of the author, it is not impossible that these quite different approaches could complete each other. In fact, the ideas in this paper could lead to a preferred $f(R)$ theory or to some subclass of such theories. Furthermore, and perhaps more likely, they could lead to some larger class of Lagrangians that is not presently included in the $f(R)$ approach.

**Funding:** This research received no external funding.

**Conflicts of Interest:** The author declares no conflict of interest.

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
