# Peer review of "Natural Lagrangians"

_universe, doi:10.3390/universe7030074_

Round 1

Reviewer 1 Report

The author considers a new representation of the general relativity by using a Lagrangian inspired by the canonical ensembles with partition functions and free energies in the statistical mechanics. As with the energy in the statistical mechanics, the author introduces the new conserved energy according to the curvature in the curved spacetime and obtains a homogeneous, isometric and closed solution which would represents an accelerated expansion of the Universe. I find this work interesting but I recommend the author to consider following points:
i) It seems that the figure 2 is not cited in the present manuscript. The author may add the citation of the figure 2.
ii) The author may add a horizontal axis label as $t$ in the figure 1, and the axes labels as $t$ and "Energy" in the figure 2 for readers' convenience.
iii) The author may correct the equation (7.7) by adding an equal sign. 
iv) Using the equation of motion (7.7), the author obtains an expanding universe solution shown in the figure 1. Here the Friedmann‐Lemaitre‐Robertson-Walker metric represents such an expanding universe and is obtained by considering the cosmological constant and the matter field associated with the energy-momentum tensor. Then, why can the author obtain such an expanding solution without the cosmological constant and the energy-momentum tensor? Do the parameters $\beta$ and $\lambda$ play a role like such the constant and the matter field? Then the author may show the values of $\beta$ and $\lambda$ with initial conditions for the solution shown in the figure 1, and discuss the derivation of this solution in detail.

Author Response

Please see the pdf-file.

Reviewer 2 Report

The paper is a first step towards a redefinition of the natural action in General Relativity. 

The proposal is an integral action containing a term depending on the scalar curvature (but raised to the 2nd power, unlike classical Hilbert Einstein action) and a term expressing the probabilistic density of the states of the metric. 

The work’s purpose, by admission of the Author himself,  is to raise the question and the critical issues about the problem without suggesting a definitive answer. In this sense my opinion is that it can be reconsidered form publication, once the following issues are resolved:

  • the heuristic argument of Section 4, regarding the vacuum case, should probably be provided with more details and calculations;
  • Remark 2 discussion should be adequately developed to discuss spacetime singularities in this context;
  • the cosmological example from Section 7 should be clarified a little. In particular: is the second term in eq (7.6) corresponding to the \log\Omega term? I would also provide the exact equations in this particular case;
  • in view of the suggested action, possible connections with f(R) gravity theories should be discussed with reference to appropriate literature on the subject.

Round 2

Reviewer 2 Report

I have read the amended version and the Author's response, I find it quite honest and I appreciate the pioneering spirit of the paper. I would only suggest to replace ref. 2 with a couple of reviews on the subject such as

S Capozziello, M De Laurentis, Extended Theories of Gravity, Physics Reports, 509, Issues 4–5, (2011) 167-321 

and

S Nojiri, S D Odintsov,
Unified cosmic history in modified gravity: From F(R) theory to Lorentz non-invariant models, Physics Reports, 505, Issues 2–4, (2011) 59-144
